# Lighting Direction Affects Leaf Morphology, Stomatal Characteristics, and Physiology of Head Lettuce (*Lactuca sativa* L.)

**DOI:** 10.3390/ijms22063157

**Published:** 2021-03-19

**Authors:** Mengzhao Wang, Hao Wei, Byoung Ryong Jeong

**Affiliations:** 1Department of Horticulture, Division of Applied Life Science (BK21 Program), Graduate School of Gyeongsang National University, Jinju 52828, Korea; meng.zhao@163.com (M.W.); oahiew@gmail.com (H.W.); 2Institute of Agriculture & Life Science, Gyeongsang National University, Jinju 52828, Korea; 3Research Institute of Life Science, Gyeongsang National University, Jinju 52828, Korea

**Keywords:** carbohydrate, cell elongation, chlorophyll fluorescence, leaf morphogenesis, phototropism, quantum yield

## Abstract

Plants are exposed to numerous biotic and abiotic stresses, and light is one of the most important factors that influences the plant morphology. This study was carried out to examine how the lighting direction affected the plant morphology by investigating the growth parameters, epidermal cell elongation, stomatal properties, and physiological changes. Seedlings of two head lettuce (*Lactuca sativa* L.) cultivars, Caesar Green and Polla, were subjected to a 12 h photoperiod with a 300 μmol·m^−2^·s^−1^ photosynthetic photon flux density (PPFD) provided by light emitting diodes (LEDs) from three directions: the top, side, and bottom, relative to the plants. Compared with the top and side lighting, the bottom lighting increased the leaf angle and canopy by stimulating the epidermal cell elongation in leaf midrib, reduced the leaf number and root biomass, and induced large stomata with a low density, which is associated with reduced stomatal conductance and carbohydrate contents. However, the proline content and quantum yield exhibited no significant differences with the different lighting directions in both cultivars, which implies that the plants were under normal physiological conditions. In a conclusion, the lighting direction had a profound effect on the morphological characteristics of lettuce, where the plants adapted to the changing lighting environments.

## 1. Introduction

Plants are continuously bombarded by biotic and abiotic signals from their environment. Being sessile and photoautotrophic, plants are stationary and cannot move away from sources of stresses, nor can they seek out a location with optimal environmental conditions [1]. Instead, they must change their developmental patterns to adapt to the environment for survival and reproduction. Many species have evolved sophisticated photosensory systems, enabling them to respond appropriately [2]. Light is one of the most important environmental cues that a plant’s developmental patterns are based on, and plants are particularly sensitive to this crucial external signal [3,4]. The higher plants indeed have evolved an elegant ability that controls the plant form according to the ambient light conditions, which is generally termed photomorphogenesis [5].

Light intensity has a profound influence on the plant morphology. Seedlings grown in dark conditions exhibit etiolated growth, characterized by small and closed cotyledons with undifferentiated chloroplasts and long hypocotyls; in contrast, light was able to inhibit hypocotyl growth and promote cotyledon opening and expansion [3]. Feng et al. found that the plant height, hypocotyl length, and abaxial leaf petiole angle decreased, while the stem diameter was increased in soybean with the increase in the light intensity [6]. Light quality is another light condition that affects plant morphology. Light quality is usually studied in a controlled environment such as plant growth chambers, glass houses, and plant factories, where artificial light sources are used to provide light. There are many kinds of artificial lights, the most common of which are high-pressure sodium (HPS) lamps, metal halide (MH) lamps, light-emitting diodes (LEDs), and fluorescent lamps. LEDs provide a way to achieve any desired light spectrum by combining different monochromatic light chips. Each of the artificial lights described above has a specific spectrum and has different effects on the plant morphology. The plants of *Anoectochilus roxburghii* grown under a blue film exhibited a significantly greater stem diameter and leaf area compared to those plants grown under a colorless plastic film [7]. A 100% red LED treatment resulted in upward or downward leaf curling in tomato, while the combination of red and blue LEDs alleviated leaf morphological abnormalities and increased the plant biomass [8]. Under a red-light environment, directional blue light irradiation triggered epidermal cell elongation of the abaxial side, resulting in inhibition of leaf epinasty in geranium [9]. In addition to the light intensity and light quality, the photoperiod also has a significant impact on the plant morphology. It is reported that a 4 h supplementary blue light increased flower bud formation and promoted flowering [10]. Besides the morphological and developmental changes, physiological changes also occurred accordingly, including in the chlorophyll content, stomatal conductance, chlorophyll fluorescence, carbohydrate content, etc.

Lettuce (*Lactuca sativa* L.) is a major edible fresh vegetable extensively grown worldwide, due to its fast growth and commercial value [11]. It was regarded as a model plant for its sensitivity to the light quality [12,13]. Numerous studies have addressed the morphological and physiological changes of lettuce in response to different light conditions [14,15,16,17,18]. However, researchers have rarely investigated the effects of different lighting directions, especially from the bottom, on the morphological changes of plants. Thus, in this present study, we investigated how lettuce responds to the lighting direction to help fine-tune their development. Moreover, this study provides a great framework on studying leaf morphogenesis, since owing to the phototropism, the leaves exhibit different morphologies in response to lighting direction and a series of changes occur during this process, such as the change of cell elongation in leaves and the expression of plant growth regulators, genes, and proteins resulting from different lighting directions, which adds another dimension in studying leaf morphogenesis in addition to light intensity, quality, and photoperiod. We believe these can be referred to in studying leaf morphogenesis considering the space and lighting efficiency in plant production in closed environments using artificial light such as plant factories.

## 2. Results

### 2.1. Analysis of the Morphological and Growth Parameters

After 30 days of cultivation, different lighting directions had a significant impact on the plants’ morphological and growth characteristics. The leaf angle of lettuce plants was considerably increased when the light was supplied from the bottom. Specifically, the maximum leaf angles of 197.0° and 146.0° were measured with lighting from the bottom, and the minimum leaf angles of 55.7° and 56.5° were observed with lighting from the top in lettuce cultivars Caesar Green and Polla, respectively. The increased leaf angle resulted in lettuce leaves bending downward (Figure 1). Overall, lighting from the side or the bottom significantly increased the plant canopy, in comparison with lighting from the top, for both lettuce cultivars.

Table 1 presents the growth and development parameters measured after 30 days of cultivation. For Caesar Green, the bottom lighting significantly decreased the shoot height, crown diameter, shoot dry weight, leaf number, and root fresh and dry weights compared to the lighting provided from top or side directions, while the greatest shoot fresh weight was obtained with lighting from the side; the leaf width and length exhibited no significant differences in response to the three lighting directions. For Polla, there were no significant differences in the shoot height, leaf number, leaf width, and root fresh weight between plants grown with lighting from the top or from the side, but the values significantly decreased when the lighting was provided from the bottom. The plants grown with lighting from the side exhibited the highest shoot fresh and dry weights, followed by those grown with lighting from the top or the bottom. In addition, compared to lighting from the top or side, lighting from the bottom dramatically enhanced the leaf length, but resulted in reduced root fresh and dry weights in both cultivars. Moreover, the shoot to root fresh weight ratio was significantly increased with the lighting from the bottom, in comparison to that with lighting from the top or side in both cultivars.

### 2.2. Analysis of the Epidermal Cell Morphology

The lighting direction exerted a considerable influence on the morphology in the epidermal cells of leaf midribs during their development (Figure 2A–D). In the cultivar Caesar Green, the upper epidermal cell length and width were greatly promoted with lighting from the bottom. Moreover, the largest ratio of the upper and lower epidermal cell lengths was also found with lighting from the bottom. In the cultivar Polla, the epidermal cell length, but not the epidermal cell width, was dramatically enhanced with lighting from the bottom. In addition, lighting from the bottom yielded the greatest cell length to width ratio in both the upper and lower epidermis. However, there was no significant differences in the ratio of the upper and lower epidermal cell lengths in response to the three lighting directions.

### 2.3. Analysis of the Stomatal Properties

The stomatal properties of lettuce leaves were highly affected by the lighting direction (Figure 3). Lighting from the bottom significantly reduced the stomatal density in both lettuce cultivars (Figure 3G,L). Interestingly, the stomatal size in lettuce leaves presented an opposite trend in response to the lighting direction, compared to the behavior of the stomatal density. The guard cell length, width of guard cell pair, and pore length and width in lettuce leaves grown with lighting from the bottom were either the same or significantly enhanced compared to those grown with lighting from the side and the top (Figure 3H–K,M–P). The stomatal micrographs certify the observations presented above (Figure 3A–F).

### 2.4. Anaysis of the Stomotal Conductance and Quantum Yield (Fv/Fm)

The stomatal conductance in cultivar Caesar Green was higher than that in cultivar Polla, and both cultivars exhibited similar trends (Figure 4A,B) in response to the different lighting directions. The stomatal conductance was significantly decreased with lighting from the bottom, and there were no significant differences in the stomatal conductance with lighting from the top and the side. The quantum yield (Fv/Fm) of plants grown with the different lighting directions was all around 0.83, and there were no differences between the two cultivars nor with the different lighting directions (Figure 4C,D).

### 2.5. Analysis of the Proline Content

The proline content in lettuce in response to the different lighting directions is presented in Figure 5. It was found that the proline contents in cultivar Caesar Green was higher than those in cultivar Polla. However, there were no significant differences in the proline content in each cultivar in response to the different lighting directions.

### 2.6. Analysis of the Chlorophyll Content

The chlorophyll content in leaves of lettuce presented interesting results. When the chlorophyll content was calculated with fresh weight, the chlorophyll a and b contents and the ratio of chlorophyll a to b exhibited no significant differences in response to the different lighting directions for both lettuce cultivars (Figure 6A,C). However, the chlorophyll a and b contents per cm^−2^ leaf area were found to be significantly reduced in both lettuce cultivars when lighting was provided from the bottom (Figure 6B,D).

### 2.7. Analysis of Carbohydrates and Soluble Proteins

The lighting direction affected the accumulation of soluble proteins, starch, and soluble sugars in both cultivars (Figure 7). For Caesar Green, the content of soluble proteins was the lowest in plants grown with lighting from the top, while the plants grown with side and bottom lighting displayed no significant differences and had a higher level of soluble proteins. The highest starch and soluble sugar contents were found in plants grown with lighting from the top, which noticeably decreased when the lighting was provided from the side and bottom. For Polla, the content of soluble proteins showed a similar result as that for Caesar Green, where the plants grown with side and bottom lighting exhibited no significant differences and higher soluble protein contents compared to plants grown with top lighting. The starch content in the plants grown with top lighting was 21.7 mg·g^−1^ FW, which was higher than that for plants grown with side and bottom lighting. The soluble sugar contents in lettuce Polla were affected by the lighting direction in a similar manner as they were in Caesar Green.

### 2.8. Expression of Photosysthesis-Related Genes

The lighting direction influenced *PsaA* and *PsbA* genes (Figure 8). The expression of *PsaA* had no significant difference between different lighting directions in Caesar Green, but was enhanced by the side lighting in Polla. The expression level of *PsbA* in Caesar Green was 2.32- and 1.25-fold higher in the side and bottom lighting, respectively, as compared to that in the top lighting, while the expression level of *PsbA* in in Polla in the side lighting increased to 2.16-fold and decreased to 0.62-fold as compared to that in the top lighting.

## 3. Discussion

Plant growth and development is highly plastic, which allows plants to adapt to a changing environment [19]. In this study, plants underwent profound changes, at the cellular level to the whole plant level, in order to adapt to the different lighting directions. With lighting from the bottom of plants, the upper epidermal cells in midribs were stimulated, elongating greater than the lower epidermal cells did (Figure 2), resulting in the increase of leaf angle and bent leaves toward the light source to capture and more efficiently use the available light (Figure 1). As a consequence, the plant height was reduced but the canopy diameter was remarkably enhanced (Table 1). In addition, the leaf length was also promoted by bottom lighting. The largest shoot fresh and dry weights were obtained in plants grown with side lighting, which was consistent with the results of a study that observed that in vitro micropropagated potato plantlets grown with a sideward lighting system had significantly shortened stems but increased dry weight and leaf area compared to those grown with downward (overhead) lighting [20]. Interestingly, the lighting direction did not only affect the leaf morphology, but also had a significant influence on the root biomass in this study. The root fresh and dry weights in lettuce were dramatically lower with side and bottom lighting compared to those with top lighting. A significant increase of the shoot to root ratio was observed with top lighting, compared to that with side and bottom lighting (Table 1). This could be explained by the negative phototropism of plant roots first discovered by Darwin [21]. There are 14 photoreceptors expressed in *Arabidopsis* plants, and most of them are also present in the roots [22,23,24]. Light is an important environmental factor for roots; a short (10-s) but strong (82 μmol·m^−2^·s^−1^ photon flux) blue light illumination of *Arabidopsis* roots could result in a remarkable increase of the root growth rate due to the immediate burst of the reactive oxygen species (ROS) [25,26]. It was speculated that the roots were more likely exposed to light with the top lighting, and light penetrated less when provided from the side and bottom of plants, and the root growth and development were affected accordingly.

The stomata are a vital structure for photosynthesis, and the stomatal density and size are regarded as indicators of plants’ acclimation and adaptation to contrasting environments [27,28]. In addition to decreasing the risk of stomatal injury from various stresses with increased stomatal density, having highly dense but small stomata is the best strategy for obtaining the highest stomatal conductance at low CO_2_ concentrations. In this study, the lettuce grown with top lighting exhibited a high density of small stomata, while the lettuce grown with bottom lighting had a lower number of, but bigger, stomata (Figure 3). Thereby, stress resistance may be weakened in plants grown with lighting from the bottom. In spite of a longer stomatal pore length, the decreased stomatal density of lettuce in response to bottom lighting resulted in the lowest stomatal conductance (Figure 4A,B).

Chlorophyll, an essential component of the light-harvesting complex, plays a crucial role in capturing and transferring photons to the reaction center of the photosystem in the primary reaction [29]. The chlorophyll can be degraded by abiotic or biotic stresses such as heat, chilling, and drought [30,31,32]. In the current study, the chlorophyll a and b contents were not significantly different in response to the different lighting directions, when calculated on the basis of the fresh leaf weight (Figure 6A,C). However, the chlorophyll a and b contents in the leaves of lettuce grown with bottom lighting decreased significantly compared to those grown with top and side lighting when calculated on the basis of the fresh leaf area (Figure 6B,D). This may be due to the greater leaf thickness resulting from the bottom lighting, such that the contents of chlorophyll were lower per unit area.

Proline accumulation occurs in plants when they are exposed to various stresses; it is believed to play adaptive roles in the plant stress tolerance, and the level of proline accumulation in plants can be 100 times greater than that of the control [33,34]. Proline accumulation has been observed when plants suffered from salinity, drought, high temperature, low temperature, and UV irradiation [35,36,37,38,39]. In addition, the photon fluence density was regarded as one of the regulators of proline accumulation in osmotically stressed *Lotus corniculatus* plants [40]. In this study, we measured the proline content of leaves in the two lettuce cultivars grown with different lighting directions. Our expectations were that the bottom lighting would stimulate the proline accumulation more, compared to the side and top lightings. However, no significant differences were found among the different lighting directions in both lettuce cultivars (Figure 5). Our inference is that the different lighting directions changed the plant morphologies, and the plants adapted perfectly to the different light conditions with the changed morphologies such that they were not stressed. This was evidenced by the Fv/Fm, the maximum quantum yield of PSII photochemistry, which is used as an indicator of the stresses caused by the PSII system [41]. A plant is considered to be exposed to environmental stresses when the Fv/Fm value is below the range of 0.80–0.84. The Fv/Fm values in the leaves of the two lettuce cultivars grown with the different lighting directions were all around 0.83 (Figure 4), which implied the plants were under normal physiological conditions.

The lighting direction had an influence on the accumulation of the primary metabolites in both lettuce cultivars. Carbohydrates, including starch and soluble sugars, are a photosynthetic product and a substrate consumed by respiration, and the carbohydrate accumulation plays an important role in the plant growth, development, and morphology [42]. Our data showed that the side and bottom lighting enhanced the soluble protein levels but reduced the starch and soluble sugar contents (Figure 7), which was a combined effect of the stomatal properties, chlorophyll contents, and use efficiency of light given in different directions.

Photosynthesis is one of the most important chemical reactions in plants, and PSI and PSII are two light energy-driven systems that synergistically function in primary energy conversion reactions [43]. The *PsaA* and *PsbA* genes encode the P700 apoproteins of PSI and the D1 protein of PSII, respectively. They can regulate the nuclear and plastid-encoded genes [44,45]. Expression levels of *PsaA* and *PsbA* genes in the side lighting increased as compared to those in the top or bottom lighting in both cultivars. Therefore, the slight improvement of shoot biomass of the plants under the side lighting may be due to the upregulation of the *PsaA* and *PsbA* genes.

## 4. Materials and Methods

### 4.1. Plant Materials and Experimental Setup

Two lettuce cultivars, Caesar Green and Polla (Asia Seed Co., Ltd., Seoul, Korea), were chosen for the examination of the phenotypic responses to the lighting direction. The Caesar Green is not heading type and has smooth leaf edges, whereas the Polla is a heading cultivar and has jagged irregular edges. The seeds were sown in 72-cell trays containing a commercial BVB (Bas Van Buuren Substrates, EN-12580, De Lier, The Netherlands) medium for germination on 26 July, 2020. When the seedlings had developed 5 true leaves, they were transplanted to 220 mL pots filled with the BVB medium. The transplanted seedlings were randomly divided into 9 groups and transferred into 3 separate plant growth chambers (Figure 9A) with a 25 °C temperature and 80% relative humidity; each group contained 12 plants (6 plants each cultivar). Each chamber was divided into 3 light-tight compartments using plates, and every plate contained 1 group of plants with 13 cm intervals between the plants. The 3 chambers were used as 3 repetitions with the same setup. In addition, 2 layers of non-woven fabric curtain was used between plates to prevent light from interacting with each other. We used LED lamps (custom made, SungKwang LED Co., Ltd., Incheon, Korea) shedding a wide spectrum ranging from 400 to 720 nm with a distinct peak (at 435 nm) in blue to supply light, and 2 modular type LED lamps were fixed 10 cm away from the top level of plants from the top, side, or bottom relative to the plants (Figure 9B–D). The light intensity was set at 300 μmol·m^−2^·s^−1^ photosynthetic photon flux density (PPFD) from 06:00 to 18:00 by adjusting the dimmer. The light intensity was measured with a quantum radiation probe (FLA 623 PS, ALMEMO, Holzkirchen, Germany) at the top-leaf-level of plant.

The plants were watered every day at 09:00 a.m. from 20 August to 19 September, 2020 with a nutrient solution with the composition as follows (in mg per L): 708.0 Ca(NO_3_)_2_·4H_2_O, 246.0 MgSO_4_·7H_2_O, 505.0 KNO_3_, 230.0 NH_4_H_2_PO_4_, 1.24 H_3_BO_3_, 0.12 CuSO_4_·5H_2_O, 4.00 Fe-ethylene diamine tetraacetic acid, 2.20 MnSO_4_·4H_2_O, 0.08 H_2_MoO_4_, and 1.15 ZnSO_4_·7H_2_O. The growth parameters were measured, and the plants were harvested and placed in liquid N_2_ in a −80 °C refrigerator for physiological analyses after 30 days of cultivation. For measuring the growth parameters, whole plants were harvested, and roots were washed carefully using tap water and cut from the shoot. The shoot height and fresh weight, number, length, and width of leaves, and root length and fresh weight were measured directly. The dry weights of shoots and roots were measured after drying for 72 h at 60 °C in an oven. The plant height was measured as the height from the crown to the top of the plant, the leaf angle was measured as the angle between the tangent to the end of the leaves and the vertical line, and the shoot to root ratio was calculated as shoot fresh weight divided by the root fresh weight.

### 4.2. Epidermal Cell and Stomata Characteristics

Upper and lower epidermal cells of leaf midribs and abaxial surfaces of leaves were carefully excised from the fully expanded third leaves of 3 randomly selected plants at a similar position for each lighting direction, in order to respectively observe the epidermal cells and stomata. The excised samples were placed on glass slides and observed with an optical microscope (ECLIPSE Ci-L, Nikon Corporation, Tokyo, Japan). The epidermal cell and stomatal properties were analyzed by ImageJ. The stomatal density was calculated as the number of stomata divided by the area where the number of stomata was recorded. The guard cell length, width of the guard cell pair, and stomatal pore length and width were measured according to the definition of Sack and Buckley [28].

### 4.3. Stomatal Conductance and Quantum Yield (Fv/Fm)

The stomatal conductance was determined using a Decagon Leaf Porometer SC-1 (Decagon Device Inc., Pullman, WA, USA). The quantum yield was measured with a FluorPen FP 100 (Photon Systems Instruments, PSI, Drásov, Czech Republic).

### 4.4. Determination of the Proline Content

To determine the proline content, we extracted 0.2 g of homogenized fresh leaf samples in 5 mL 3% sulfosalicylic acid solution for 20 min at 100 °C. Then, 2 mL of the proline extract, 2 mL acetic acid, and 2 mL acid ninhydrin solution were mixed and incubated at 100 °C for 30 min. After cooling down to room temperature, 4 mL of toluene was added, and the mixture was vortex-oscillated for 1 min. The absorbance of the upper layer solution at 520 nm was recorded with a UV spectrophotometer (Libra S22, Biochrom Ltd., Cambridge, UK).

### 4.5. Chlorophyll Analyses

In this study, we used 2 sampling methods to determine the chlorophyll content. The first sampling method used 0.1 g samples of fresh leaves, and the second sampling method used 2 cut leaf discs using a puncher (1.2 cm in diameter). Both samples used the same protocol for measuring the chlorophyll a and b as described in a previous study [46]. The contents of chlorophyll a and b were determined using the following formulae:(1)$$\mathrm{Chlorophyll}\;a=\frac{\left(12.72\times \mathrm{OD}\mathrm{at}663\;\mathrm{nm}-2.59\times \mathrm{OD}\mathrm{at}645\;\mathrm{nm}\right)\times V}{\mathrm{Leaf fresh weight or area}}$$
(2)$$\mathrm{Chlorophyll}\;b=\frac{\left(22.88\times \mathrm{OD}\mathrm{at}645\;\mathrm{nm}-4.67\times \mathrm{OD}\mathrm{at}663\;\mathrm{nm}\right)\times V}{\mathrm{Leaf fresh weight or area}}$$

The chlorophyll content was expressed as milligram of chlorophyll per gram of fresh leaf weight and milligram of chlorophyll per square centimeter of fresh leaf area.

### 4.6. Determination of the Carbohydrate and Soluble Protein Contents

The contents of starch and soluble sugars were determined by the Anthrone colorimetric method according to Ren et al. [47]. Soluble proteins were extracted with a sodium phosphate buffer and measured colorimetrically according to the protocol described by Muneer et al. [48].

### 4.7. Quantitative Real-Time PCR Analysis

Total RNA was extracted from leaves using an Easy-Spin total RNA extraction kit (iNtRON Biotechnology, Seoul, Korea), then used for first-stand cDNA synthesis by the GoScript Reverse Transcription System (Promega, Madison, WI, USA) according to the manufacturer’s protocols. The gene expression level was determined using a real-time PCR system (CFX96, Bio-Rad, Hercules, CA). Reaction volumes (20 µL) contained 1 µL of cDNA, 1 µL of each amplification primer (10 µM), 10 µL of 2 × AMPIGENE qPCR Green Mix Lo-ROX (Enzo Life Sciences Inc., Farmingdale, NY, USA), and 7 µL ddH_2_O (double distilled water). The 2^−ΔΔCt^ method was conducted to determine the relative levels of gene expressions, and the 18S gene was used as the reference gene. The primers used are listed in Table 2.

### 4.8. Data Collection and Analysis

The experimental assays used to obtain all results were repeated 3 times and are presented as the mean ± standard error. Significant differences among the treatments were assessed by analysis of variance (ANOVA) followed by Duncan’s multiple range test at *p* < 0.05 with the SAS (Statistical Analysis System, V. 9.1, Cary, NC, USA) program.

## 5. Conclusions

This study demonstrated that the lighting direction affected the lettuce morphology by regulating the epidermal cell elongation, stomatal density and pore size, contents of chlorophyll and carbohydrates, and expression of *PsaA* and *PsbA*. Specifically, the lighting from bottom stimulated the upper epidermal cell elongation in leaf midrib, enhanced leaf angle and plant canopy, reduced leaf number and root biomass, induced large stomata with a low density, and decreased stomatal conductance and carbohydrate contents, while the proline content and quantum yield had no differences between lighting directions. Correspondingly, the changed morphology adapted lettuce to the light from different directions and helped them function normally. Further investigation remains to study the internal structural change of leaves and how the lighting direction affects epidermal cell elongation.

## Figures and Tables

**Figure 1 ijms-22-03157-f001:**
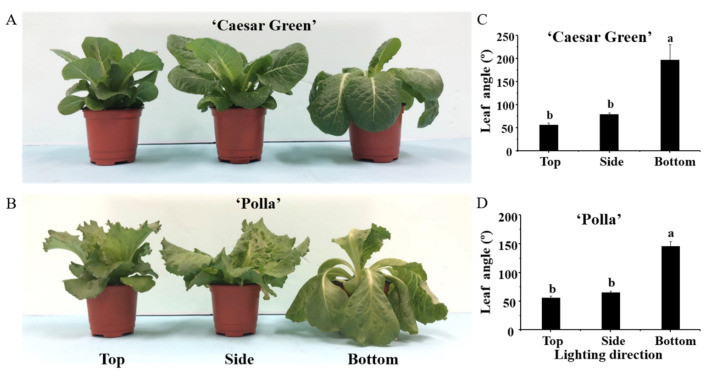
Morphology (**A**,**B**) and leaf angles (**C**,**D**) of lettuce Caesar Green and Polla as affected by the lighting direction after 30 days of cultivation. Vertical bars indicate the means ± stand error (*n* = 3). Different small letters indicate the significant separation within treatments by the Duncan’s multiple range test at *p* ≤ 0.05.

**Figure 2 ijms-22-03157-f002:**
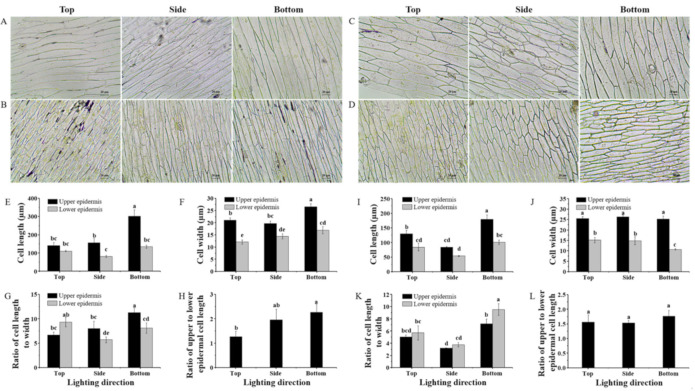
Upper and lower epidermal cell morphology of lettuce Caesar Green (**A**,**B**) and Polla (**C**,**D**), cell length and width, ratio of cell length to width, and ratio of the upper and lower epidermal cell lengths of lettuce Caesar Green (**E**–**H**) and Polla (**I**–**L**) as affected by the lighting direction. Vertical bars indicate the means ± stand error (*n* = 3). Different small letters indicate the significant separation within treatments by the Duncan’s multiple range test at *p* ≤ 0.05. Bars indicate 20 μm.

**Figure 3 ijms-22-03157-f003:**
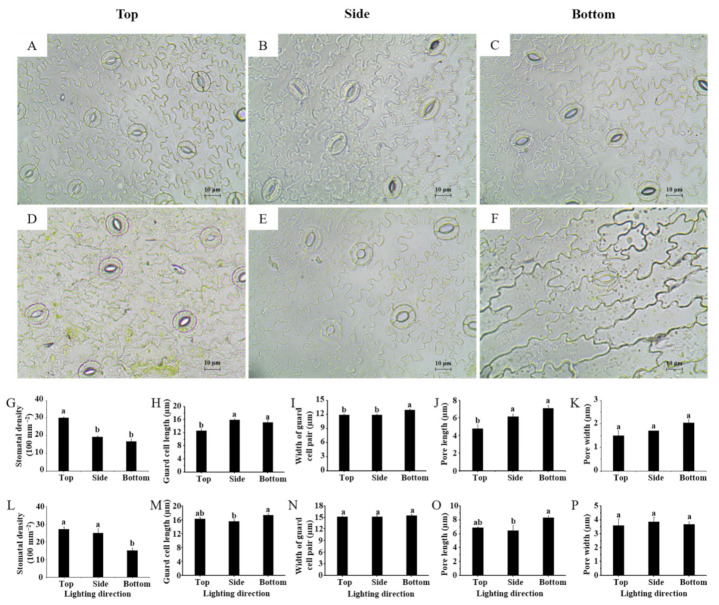
Stomatal micrographs of lettuce Caesar Green (**A**–**C**) and Polla (**D**–**F**) leaves, and stomatal density, guard cell length, width of guard cell pair, pore length and width of lettuce Caesar Green (**G**–**K**) and Polla (**L**–**P**). Vertical bars indicate the means ± stand error (*n* = 3). Different small letters indicate the significant separation within treatments by the Duncan’s multiple range test at *p* ≤ 0.05. Bars indicate 10 μm.

**Figure 4 ijms-22-03157-f004:**
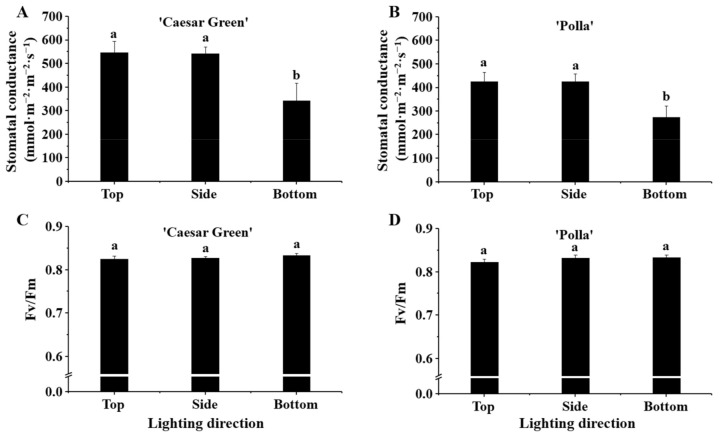
The effects of the lighting direction on the stomatal conductance and Fv/Fm of two lettuce cultivars Caesar Green (**A**,**B**) and Polla (**C**,**D**). Vertical bars indicate the means ± stand error (*n* = 3). Different small letters indicate the significant separation within treatments by the Duncan’s multiple range test at *p* ≤ 0.05.

**Figure 5 ijms-22-03157-f005:**
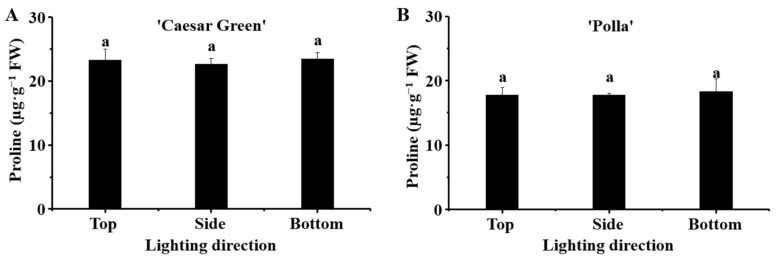
Effect of the lighting direction on the proline content in leaves of lettuce Caesar Green (**A**) and Polla (**B**). Vertical bars indicate the means ± stand error (*n* = 3). Different small letters indicate the significant separation within treatments by the Duncan’s multiple range test at *p* ≤ 0.05.

**Figure 6 ijms-22-03157-f006:**
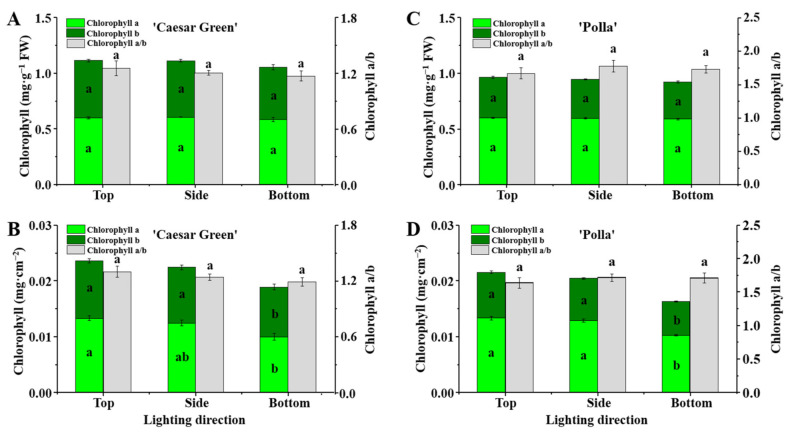
The effects of the lighting direction on contents of chlorophylls a and b, and chlorophyll a to b ratio in leaves of lettuce Caesar Green and Polla calculated on the basis of fresh leaf weight (**A**,**C**) and fresh leaf area (**B**,**D**). Vertical bars are the means ± stand error (*n* = 3). Different small letters indicate the significant separation within treatments by the Duncan’s multiple range test at *p* ≤ 0.05.

**Figure 7 ijms-22-03157-f007:**
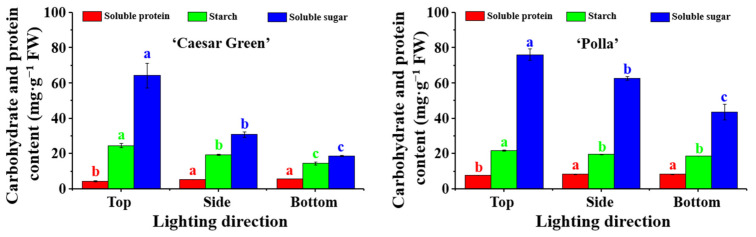
Effect of the lighting direction on the carbohydrate and protein contents in leaves of lettuce Caesar Green and Polla. Vertical bars indicate the means ± stand error (*n* = 3). Different small letters indicate the significant separation within treatments by the Duncan’s multiple range test at *p* ≤ 0.05.

**Figure 8 ijms-22-03157-f008:**
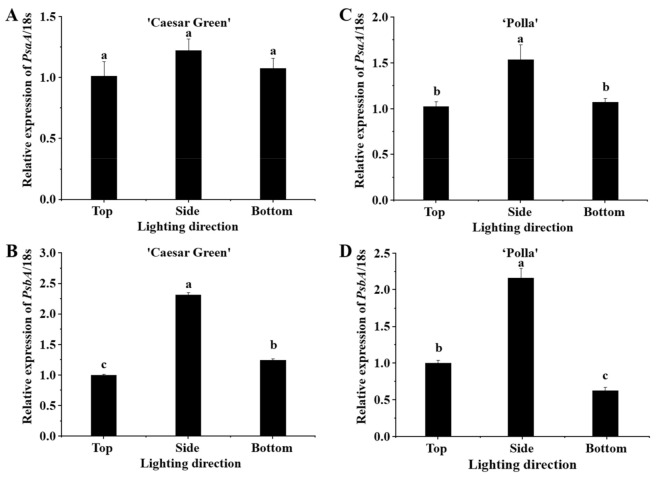
Relative expression levels of *PsaA* and *PsbA* in lettuce Caesar Green (**A**,**B**) and Polla (**C**,**D**). Vertical bars indicate the means ± stand error (*n* = 3). Different small letters indicate the significant separation within treatments by the Duncan’s multiple range test at *p* ≤ 0.05.

**Figure 9 ijms-22-03157-f009:**
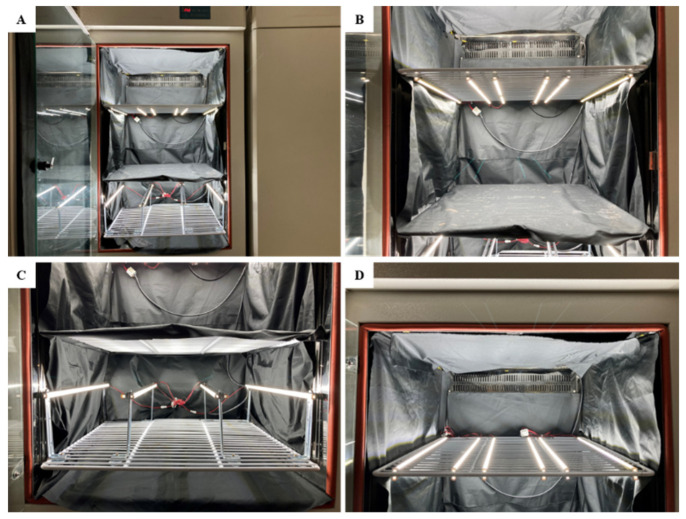
The plant growth chamber used in the study (**A**) and the light-emitting diode (LED) installed from the top (**B**), side (**C**), and bottom (**D**) relative to the plants.

**Table 1 ijms-22-03157-t001:** Influence of the lighting direction on the growth and development of lettuce grown for 30 days.

Cultivar (A)	Lighting Direction (B)	Shoot					Leaf			Root			Shoot/Root Ratio (Fresh Weight)
		Height (cm)	Crown Diameter (mm)	Canopy Diameter (cm)	Fresh Weight (g)	Dry Weight (g)	Number	Length (cm)	Width (cm)	Length (cm)	Fresh Weight (g)	Dry Weight (g)	
“Caesar Green”	Top	16.0 a ^z^	7.8 a	15.7 cd	19.34 b	1.28 ab	21 a	11.5 c	6.3 c	20.3 b	6.74 a	0.43 a	2.96 c
	Side	15.4 ab	7.2 ab	19.2 ab	20.50 ab	1.24 ab	20 a	13.2 a-c	6.9 c	21.8 b	5.39 ab	0.28 b	4.10 bc
	Bottom	14.2 c	6.1 b	19.3 ab	19.21 b	1.05 b	17 b	14.8 a	6.3 c	21.1 b	3.82 bc	0.22 bc	5.72 b
“Polla”	Top	15.8 ab	7.2 ab	15.1 d	20.39 ab	1.28 ab	16 b	11.8 c	9.8 a	26.4 a	4.83 b	0.32 ab	4.59 bc
	Side	15.8 ab	6.5 b	17.5 bc	23.57 a	1.38 a	16 b	12.5 bc	10.0 a	23.3 ab	4.23 bc	0.28 b	5.82 b
	Bottom	14.9 bc	6.3 b	20.0 a	18.85 b	1.00 b	13 c	13.6 ab	8.7 b	23.9 ab	2.75 c	0.16 c	7.48 a
*F*-test	A	NS ^y^	NS	NS	NS	NS	***	NS	***	**	*	NS	***
	B	***	*	***	*	NS	***	***	*	NS	**	*	***
	A × B	NS	NS	NS	NS	NS	NS	NS	NS	NS	NS	NS	NS

^z^ Mean separation within columns by Duncan’s multiple range test at *p* ≤ 0.05. ^y^ NS, *, **, ***, non-significant or significant at *p* ≤ 0.05, 0.01, or 0.001, respectively.

**Table 2 ijms-22-03157-t002:** The nucleotide sequences of primers used in the present study.

Gene	Forward (5′ to 3′)	Reverse (5′ to 3′)
*PsaA*	ATTTGACTGTTGGCGGGTCT	CCCGGTCTAGCCCATTCC
*PsbA*	ATTCGTGCGCTTGGGAGTC	AAGACGGTTTTCGGTGCTG
*18S*	ATGATAACTCGACGGATCGC	CTTGGATGTGGTAGCCGT

## Data Availability

The data presented in this study are available on request from the corresponding author.

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
