# Peer review of "Lighting Direction Affects Leaf Morphology, Stomatal Characteristics, and Physiology of Head Lettuce (Lactuca sativa L.)"

_ijms, 2021, doi:10.3390/ijms22063157_

Round 1

Reviewer 1 Report

Review of the manuscript

[IJMS] Manuscript ID: ijms-1133448

“Lighting Direction Affects Leaf Morphology, Stomatal Characteristics, and Physiology of Head Lettuce (Lactuca sativa L.)”

The presented manuscript showed the reaction of plants (Lactuca sativa L. cultivars, ‘Caesar Green’ and ‘Polla’) to lightening from 3 different directions: top, side and bottom. Performed analyses concerned: epidermal cell elongation, stomatal properties, proline, chlorophyll, carbohydrates and soluble protein content. Obtained results showed that the bottom lighting increased the leaf angle, stimulated the epidermal cell elongation, reduced the leaf number and root biomass, and influenced the stomata development as well as reduced stomatal conductance and carbohydrate contents.

From the cognitive point of view, these results are interesting, but why it is important to get the knowledge of the effect of light directionality, in this case from the bottom on understanding regulation of plant growth processes is not mentioned. The influence of the light directionality on growth and physiological processes is well understood and in this respect the work only shows that light from the bottom has an effect, but this can be expected with a general knowledge of the effect of litter on leaf morphogenesis. I did not find in discussion, why such research was undertaken. Authors wrote in the introduction: Besides, this study provides a great idea on studying leaf morphogenesis, since owing to the phototropism, the leaves exhibit different morphologies in response to lighting direction and a series of changes occur during this process, which adds another dimension in studying leaf morphogenesis in addition to light intensity, quality, and photoperiod.” – and this statement is not discussed.

General comments to the manuscript

Title – OK

Abstract – OK.

Introduction – OK

Results – are well presented, some minor comments are in the PDF file.

For example: “cell of leaf veins” – I do not understand what Author mean: the vascular veins or the cell lines? the shape of pavement epidermal cells presented on fig. 2 and 3 are different – it must be explained why?

Discussion – in general it is good, but there is no discussion of the results obtained in the context of how lighting from the bottom can help to understand the mechanisms regulating plant growth; no indication what mechanisms regulating the development of the leaf can be better known by illuminating the leaves from the bottom. Thus, it appears to be superficial.

Materials and methods – all comments are in the PDF file.

In my opinion, this manuscript should not be published in the present form in The International Journal of Molecular Sciences.

All my detailed comments are marked in PDF file.

Reviewer 2 Report

The work presented for review is valuable because it brings additional knowledge to the cultivation of leaf vegetables in different directions of plant lighting. It indicates that the plant adapts quite quickly to light coming from different directions and allows them to function normally. The work can be printed after minor corrections marked in the text 

Reviewer 3 Report

The problems related to the lighting of plants using LED lamps is a very current subject. The manuscript is interesting and uses many research methods to explain the response of plants to different lighting direction using LEDs.

Line 88 – 91 shoot hight is lowest with bottom lighting compared to top and side lighting, please consider in the other descriptions of the results the significance given in Table 1 between the means and marked with different letters. This also applies to the description of the other plant traits tested.

Line 97 this statement is only true for bottom lighting

Line 99 -101 as above

Line 106 -107 means marked with the same letters are not significantly different on Duncan's test.

Line 106 Dry weight is the total of one whole plant, how the leaves and roots were prepared to obtain dry weight because this is not mentioned in the M&M. Please also describe in the methodology how the plant roots were extracted from the potting substrate, this substrate in the pots was organic?

Line 283 shoot biomass was increased with side lighting but this was not a significant difference compared to the other combinations

Line 287 please give information on the differences between the lettuce cultivars

  • In the M&M chapter, please give more details about setting up the experiment and installing the lamps
  • How many plants were in the combination, how many repetitions and how many plants per repetition
  • What is the difference between these 3 growth chambers
  • How many LED lamps and which light spectrum were used
  • Was there side lighting on both sides of the plant?
  • please clarify if the plant was lighted from below, if it used only this light source and if the intensity of 300 μmol/m2/s was measured in which position
  • It is not clear how the lamps illuminating the plants from the bottom were installed, perhaps you should make a diagram of the positioning of the lamps and plants in the chambers
  • Please give the dates for the trial set-up, sowing of seeds
  • Please give the dates of three repetitions of the presented research (if they were repeated)
  • Was this study conducted once under controlled conditions.
  • Please state what the microclimate parameters were during the 30 days of plant growth
  • Line 360 please reword the conclusions. Please state in points the conclusions obtained.
